# *Mycobacterium tuberculosis* precursor rRNA as a measure of treatment-shortening activity of drugs and regimens

Nicholas D. Walter [1,2,3,29 ✉], Sarah E. M. Born [4,29], Gregory T. Robertson[3,5,29], Matthew Reichlen[4], Christian Dide-Agossou[6], Victoria A. Ektnitphong[5], Karen Rossmassler[1,2], Michelle E. Ramey[5], Allison A. Bauman[5], Victor Ozols[1,2], Shelby C. Bearrows[1,2], Gary Schoolnik[7], Gregory Dolganov[7], Benjamin Garcia [8,9], Emmanuel Musisi[10,11], William Worodria[10], Laurence Huang[12,13,14], J. Lucian Davis [15,16], Nhung V. Nguyen[17], Hung V. Nguyen[17], Anh T. V. Nguyen[17], Ha Phan[17], Carol Wilusz[5], Brendan K. Podell[5], N' Dira Sanoussi [18], Bouke C. de Jong [19], Corinne S. Merle[20,21], Dissou Affolabi[18], Helen McIlleron[22,23], Maria Garcia-Cremades [24], Ekaterina Maidji[25], Franceen Eshun-Wilson[25], Brandon Aguilar-Rodriguez[25], Dhuvarakesh Karthikeyan[25], Khisimuzi Mdluli[26], Cathy Bansbach[27], Anne J. Lenaerts[5], Radojka M. Savic [3,12,24,28], Payam Nahid[3,12,14,17,28], Joshua J. Vásquez [3,12,14,25,28] & Martin I. Voskuil [3,4 ✉]

There is urgent need for new drug regimens that more rapidly cure tuberculosis (TB). Existing TB drugs and regimens vary in treatment-shortening activity, but the molecular basis of these differences is unclear, and no existing assay directly quantifies the ability of a drug or regimen to shorten treatment. Here, we show that drugs historically classified as sterilizing and non-sterilizing have distinct impacts on a fundamental aspect of *Mycobacterium tuberculosis* physiology: ribosomal RNA (rRNA) synthesis. In culture, in mice, and in human studies, measurement of precursor rRNA reveals that sterilizing drugs and highly effective drug regimens profoundly suppress *M. tuberculosis* rRNA synthesis, whereas non-sterilizing drugs and weaker regimens do not. The rRNA synthesis ratio provides a readout of drug effect that is orthogonal to traditional measures of bacterial burden. We propose that this metric of drug activity may accelerate the development of shorter TB regimens.

A full list of author affiliations appears at the end of the paper.

Since the advent of effective antibiotic therapy, there has been an enduring quest to shorten the length of treatment required to reliably cure tuberculosis (TB)[1–5]. Current standard regimens range in duration from 6 months for drug-susceptible TB to years for some drug-resistant TB infections[6,7]. Two factors considered crucial to treatment-shortening activity are the capacity of a drug to penetrate and accumulate in lung lesions (i.e., pharmacokinetic properties)[8,9] and the inherent activity of a drug against residual drug-tolerant *Mycobacterium tuberculosis* (*Mtb*) populations that survive initial drug killing[2,4,10,11].

Our limited understanding of drug activity against residual *Mtb* populations that withstand early killing impedes development of shorter treatments[2]. Conventionally, treatment-shortening activity has been viewed as synonymous with killing drug-tolerant *Mtb*, leading to the historical term "sterilizing activity."[12] However, it is unclear that killing alone entirely explains the ability of drugs or regimens to cure TB[1,2].

Drug activity against residual *Mtb* populations is currently not directly measurable[2]. Instead, in a lengthy and expensive process, the activity of drugs has been cataloged empirically based on the degree to which drugs shorten the time needed to achieve non-relapsing cure in animal relapse studies and human trials[2,13]. Rifampin, pyrazinamide, and bedaquiline have been classified as potent sterilizing agents[12,14]. Many other drugs, including isoniazid, streptomycin, and ethambutol, are classified as non-sterilizing because they may be bactericidal during the first days of treatment but contribute only modestly to shortening the time needed to achieve non-relapsing cure[12,14]. To develop new, shorter TB regimens, there is a critical need to measure treatment-shortening activity at an early stage of drug and regimen evaluation[1,4,10,11]. In this work, we evaluated an additional dimension of treatment-shortening activity: whether sterilizing drugs and non-sterilizing drugs have different impacts on the physiologic state of *Mtb* and whether this difference can be exploited as a biomarker for treatment-shortening potential[1,11].

Here we show that sterilizing and non-sterilizing drugs have distinct effects on rRNA synthesis, a fundamental bacterial physiologic parameter. Drugs and regimens that shorten treatment and ensure durable, non-relapsing cure profoundly suppress *Mtb* rRNA synthesis, whereas drugs and regimens with lower sterilizing activity allow surviving *Mtb* populations to sustain ongoing rRNA synthesis. Quantification of rRNA synthesis may serve as a marker of the ability of a drug or drug regimen to shorten TB treatment.

## Results

### Precursor rRNA abundance provides a measure of *Mtb* physiology.
*Mtb* synthesizes a pre-rRNA transcript that includes mature rRNA (16S and 23S) and short-lived spacer sequences [external transcribed spacer 1 (ETS1) and internal transcribed spacer 1 (ITS1)] (Fig. 1a). Since spacer regions are rapidly degraded while mature rRNA is stable, the abundance of pre-rRNA relative to mature rRNA serves as a measure of ongoing rRNA synthesis[15]. For internal normalization, we defined the rRNA synthesis (RS) ratio as the ratio of ETS1 to 23S rRNA copies (measured via droplet digital PCR) multiplied by $10^4$. We also confirmed that measurement of ITS1 recapitulates results based on ETS1 (Supplementary Figs. 1, 5 and 7).

We used three experimental models to confirm the RS ratio is a physiologic marker that correlates with growth[16,17]. First, a progressive oxygen depletion model[18] demonstrated that *Mtb* growth and rRNA synthesis decreased in parallel (Fig. 1b). Second, we evaluated the time course of rRNA synthesis in the untreated chronic murine TB model. During the first days after infection, the burden of *Mtb* colony-forming units (CFU) in the lungs increased dramatically and the RS ratio was high (Fig. 1c), consistent with rapid bacillary replication. After the onset of adaptive immunity[19,20], the *Mtb* burden plateaued and rRNA synthesis slowed (day 25). By day 53, the RS ratio was low but far from maximally suppressed, consistent with previous "replication clock" experiments showing that the plateau in CFU reflects a dynamic equilibrium between death and ongoing replication rather than a dormant non-replicating state[19,21,22].

Third, the RS ratio demonstrated that key histologic micro-environments of the C3HeB/FeJ mouse harbor distinct *Mtb* populations that have markedly different levels of ongoing rRNA synthesis. Unlike other murine models, after low-dose aerosol infection with *Mtb*, the C3HeB/FeJ mouse develops type 1 granulomas (Fig. 2a) that exhibit a large central area of hypoxic caseous necrosis surrounded by a rim of inflammatory cells that include viable and degenerate neutrophils and heavily vacuolated macrophages, and are bounded by a cuff of fibrotic compressed lung tissue and infiltrating leukocytes[9,23–26]. Using quantitative multiplexed RNA in situ hybridization (ISH), we measured the ratio of pre-rRNA to 23S rRNA signals within individual bacilli (Fig. 2a–g and Supplementary Figs. 2–4). *Mtb* were present at similar densities in the inflammatory rim and the caseum (39.3 vs 37.7 bacilli per $\mu m^2$) (Supplementary Fig. 3c). While 23S rRNA signals were similar in the rim and caseum ($P = 0.62$), the pre-rRNA mean fluorescent intensity (MFI) was significantly lower in the caseum ($P < 0.0001$), indicating a quiescent caseum *Mtb* population with decreased rRNA synthesis. Evaluation of the RS ratio on an individual-bacillary level revealed population heterogeneity in the RS ratio within both the inflammatory rim and caseum (Fig. 2e). The *Mtb* population of the rim was more heterogeneous than the population of caseum (variance = 0.64 vs. 0.38, respectively $P < 0.0001$). These observations indicate that the RS ratio measures a fundamental physiological parameter of *Mtb* populations.

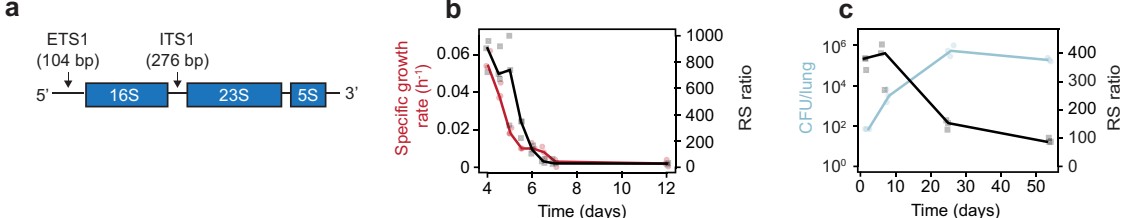

**Fig. 1 RS ratio correlates with *Mtb* replication. a** *Mtb* ribosomal operon. **b** The RS ratio (black) mirrored specific growth rate (red) in an in vitro progressive oxygen depletion model. Dots represent values from three independent experiments. Lines connect median values. **c** In the BALB/c chronic infection model, the lung *Mtb* CFU burden (blue) rose rapidly and then plateaued over time. The RS ratio (black) indicated rapid initial rRNA synthesis that slowed as *Mtb* burden plateaued. Dots represent values from three individual mice. Lines connect mean values from each timepoint. Source data are provided as a Source Data file.

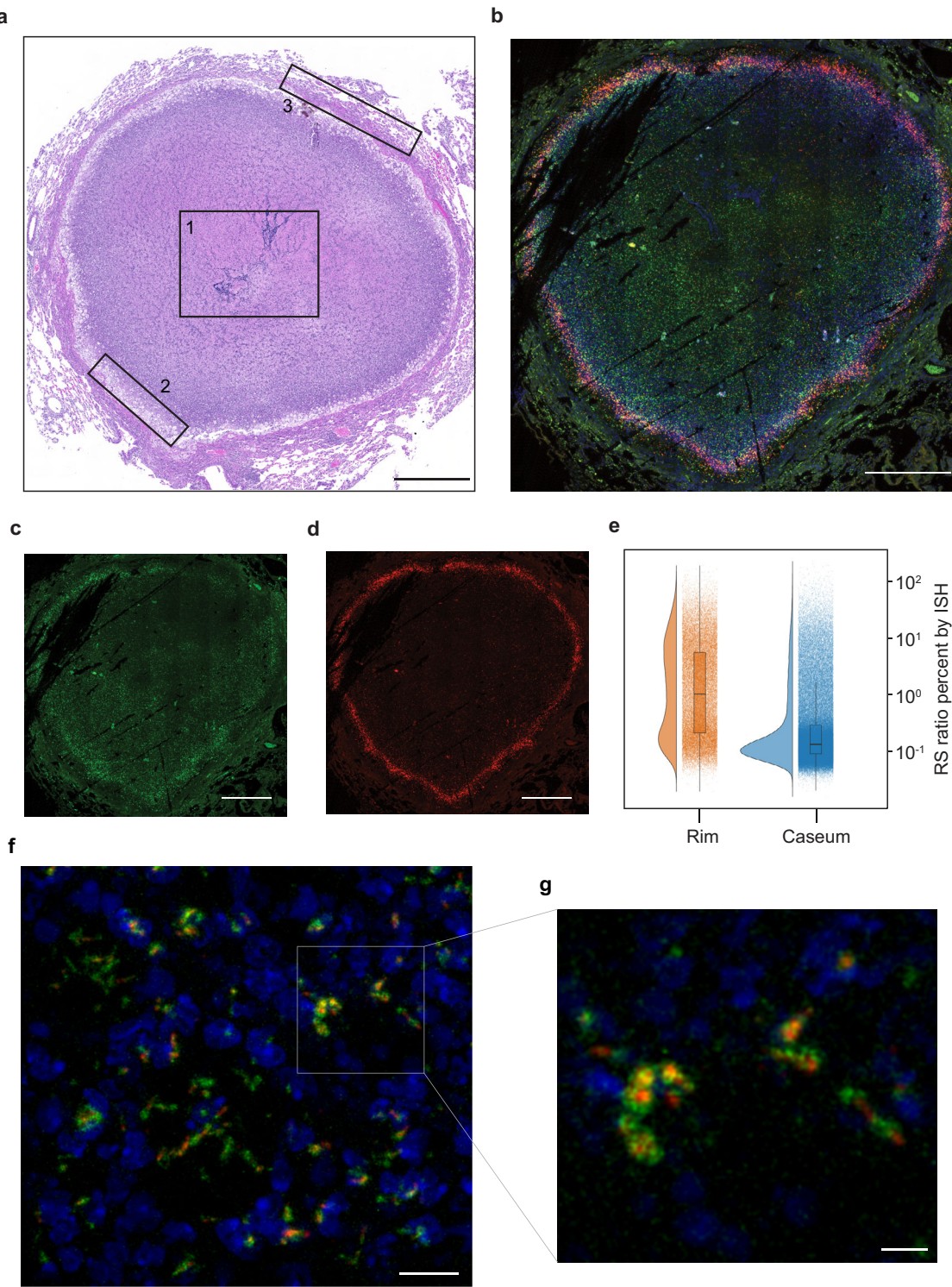

**Sterilizing potency and rRNA synthesis in vitro**. We tested canonical TB antimicrobials for effects on rRNA synthesis using in vitro time-kill experiments. Drugs with well-established sterilizing potency included rifampin, bedaquiline, and pyrazinamide. Pyrazinamide was excluded from in vitro analysis because its full activity requires in vivo conditions[27]. The well-established non-sterilizing drugs examined included isoniazid, streptomycin, and ethambutol. We used drug concentrations that maximally decreased *Mtb* CFU (up to 99% after 3 days).

Sterilizing drugs (rifampin and bedaquiline) suppressed rRNA synthesis significantly more than non-sterilizing drugs (isoniazid, streptomycin, and ethambutol) (Fig. 3a–e). Consistent with direct inhibition of RNA polymerase, rifampin decreased the RS ratio 130-fold in 6 h and 546-fold in 48 h. Bedaquiline, an inhibitor of *Mtb* ATP-synthase, also rapidly halted ongoing rRNA synthesis, decreasing the RS ratio sixfold in 6 h and 149-fold in 48 h. In contrast, the three non-sterilizing drugs had a limited impact on rRNA synthesis. With isoniazid, the RS ratio rose 1.4-fold in 6 h

**Fig. 2 Single-bacillus RS ratio in situ indicates that *Mtb* rRNA synthesis is lower in caseum than the granuloma's inflammatory rim. a** H&E-stained section of a single typical lung granuloma from C3HeB/FeJ mouse. Most of the lesion is comprised of caseous necrosis (1). The caseum is surrounded by a rim of inflammatory cells that include viable and degenerate neutrophils and heavily vacuolated macrophages (2). The granuloma is contained by an outer rim of compressed lung tissue, fibrosis, and infiltrating leukocytes (3). **b** Granuloma from C3HeB/FeJ mouse lung with multiplexed ISH overlay staining for 23S rRNA (green), pre-rRNA (red), and DAPI for host-cell nuclei (blue). The channels for ISH are shown individually in **c** and **d**. **c** 23S rRNA ISH identified *Mtb* present throughout the granuloma. **d** pre-rRNA ISH indicated lower *Mtb* rRNA synthesis in the caseum compared with the inflammatory rim. **e** Graphical analysis as well as statistical testing of the RS ratio by ISH showed that there was greater population heterogeneity in rRNA synthesis in the inflammatory rim (orange) than in the caseum (blue). Components of this raincloud plot are: (1) density plots for the distribution of 164,878 RS ratio values for individual bacilli in a single granuloma on a $\log_{10}$ scale, (2) scatterplots to visualize all points measured, and (3) boxplots to present the range of values in the RS ratio. The center and ends of the box represent the median and first and third quartiles of the RS ratio. The boxplot whiskers represent the maximum and minimum values in each group. **f** Magnification of high-powered images depicts co-occurrence of rRNA signals within individual bacilli. **g** Further magnification demonstrates 23S signals distributed in a reticular pattern around a central confluence of pre-rRNA signals. Panels **a–d** were imaged at ×40; scale bars represent 500 μm. Z-stacked images surrounding this image are provided in Supplementary Fig 4. Panels **f** and **g** were imaged at ×63; scale bars represent 10 and 5 μm, respectively. Panels **a–g** are results from a single lung granuloma. Replicate results from two additional granulomas from two different mice in separate independent experiments are provided in Supplementary Fig. 2.

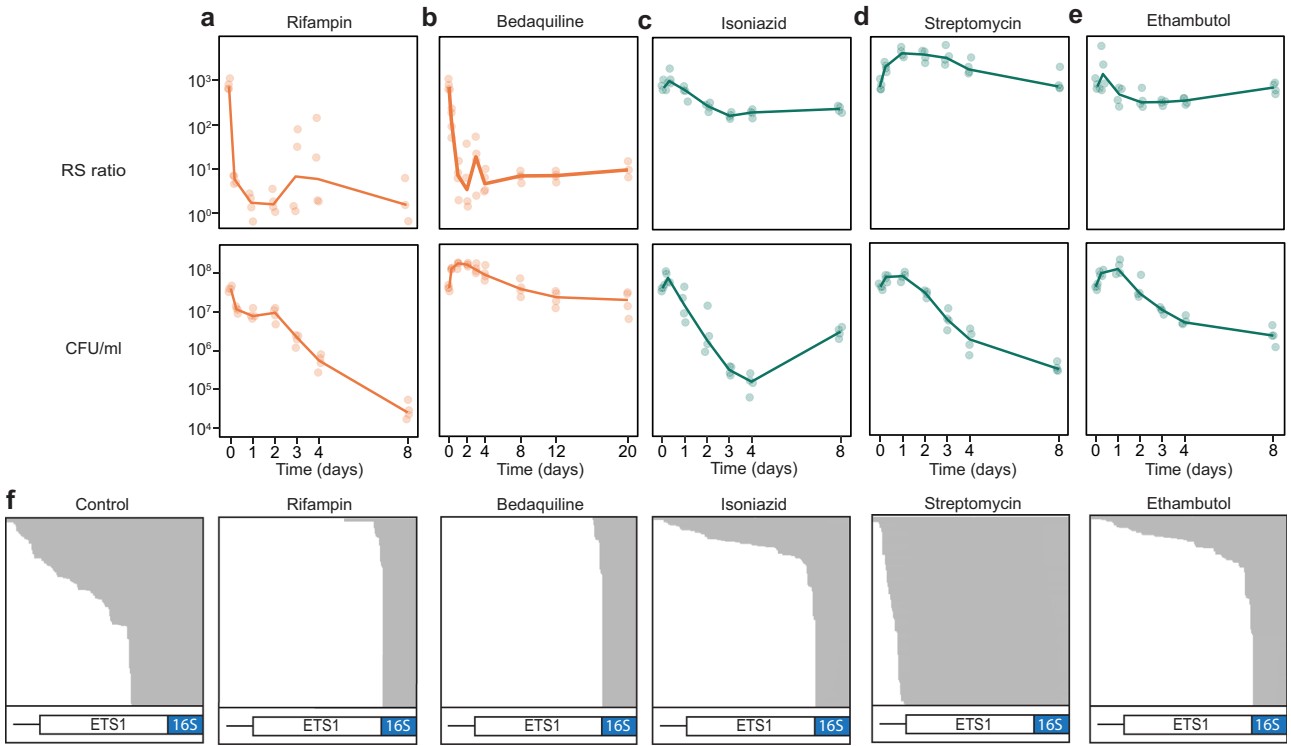

**Fig. 3 Canonical sterilizing drugs suppress rRNA synthesis more than non-sterilizing drugs in vitro.** Change in RS ratio and CFU burden in time-kill experiments with **a** rifampin, **b** bedaquiline, **c** isoniazid, **d** streptomycin, and **e** ethambutol. Sterilizing and non-sterilizing drugs shown in orange and green, respectively. Dots represent values from three independent experiments. Lines connect mean values from each timepoint. **f** RNAseq results for log-phase untreated control and after 48 h of drug exposure. Integrative Genomics Viewer (IGV) images demonstrate abundant ETS1 reads in the control sample compared with near-absence of ETS1 reads after exposure to rifampin and bedaquiline. Gray shading denotes individual reads. White and blue bars indicate the ETS1 pre-rRNA and 16S rRNA regions, respectively. Source data for a–e are provided as a Source Data file.

and then decreased 2.9-fold in 48 h. With streptomycin, the RS ratio rose 2.1-fold in 6 h and 5.2-fold in 48 h. With ethambutol, the RS ratio rose 1.6-fold in 6 h then decreased 2.0-fold in 48 h. At all timepoints, RS ratios for bedaquiline and rifampin were significantly lower than for any of the non-sterilizing drug (highest *P* value among all comparisons was 0.004, Supplementary Table 1).

RNAseq results confirmed that sterilizing drugs reduce pre-rRNA abundance far more than non-sterilizing drugs (Fig. 3f). After 48 h of exposure, ETS1 was substantially reduced with rifampin or bedaquiline relative to isoniazid, streptomycin, or ethambutol. To confirm drug effect on rRNA synthesis via an

alternative approach, we used radionucleotide incorporation to quantify de novo nucleic acid synthesis. Rifampin and bedaquiline suppressed synthesis of DNA and total RNA (comprised primarily of rRNA) markedly more than isoniazid, streptomycin, or ethambutol (Supplementary Fig. 6b).

The effect of drugs on the RS ratio was independent of their effect on CFU. Bedaquiline immediately inhibited rRNA synthesis while CFU did not decrease appreciably for >8 days. In contrast, rifampin decreased the RS ratio and CFU rapidly and simultaneously. The three non-sterilizing drugs decreased CFU quickly but had modest effects on the RS ratio. As anticipated, isoniazid monotherapy developed resistance but this resistant

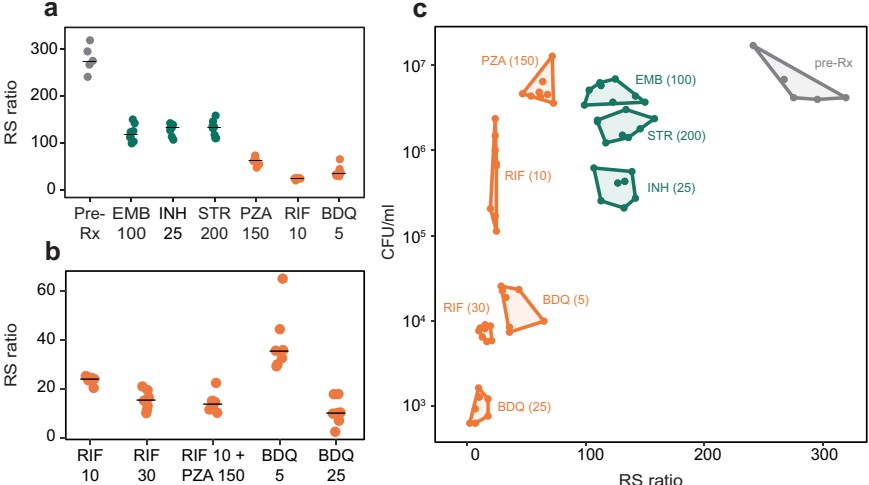

**Fig. 4 Canonical sterilizing drugs suppress rRNA synthesis more than non-sterilizing drugs in mice. a** RS ratio in lung tissue of BALB/c mice treated for 28 days with individual non-sterilizing (green) and sterilizing (orange) drugs. Dots indicate individual mice ($N = 8$ per treatment). Horizontal lines indicate group means. **b** Dose–response relationship between rifampin and bedaquiline dose and the RS ratio. Also shown is the combination of rifampin 10 mg kg$^{-1}$ and pyrazinamide. Dots indicate individual mice ($N = 8$ per treatment). Horizontal lines indicate group means. **c** Contrasting effects of 28-day treatment with individual drugs/doses on CFU and the RS ratio in BALB/c mice. A convex hull was used to display data groupings. Source data are provided as a Source Data file.

population did not expand sufficiently to have a substantial impact on the RS ratio until well after 2 days of drug exposure (Supplementary Fig. 5i).

**rRNA synthesis and sterilizing potency of individual drugs in mice.** We tested the effect of individual sterilizing and non-sterilizing drugs in the BALB/c mouse high-dose aerosol efficacy model. One day after infection, the *Mtb* burden in the lungs was low (3.8 log$_{10}$ CFU lung$^{-1}$) and the RS ratio was high (median: 290, interquartile range (IQR): 285–360), indicating rapid rRNA synthesis and bacillary replication. At the start of treatment (day 11), the *Mtb* lung burden had increased 1000-fold and the RS ratio remained high (median: 275, IQR: 267–295).

After 28 days of treatment at standard doses meant to mimic human drug exposures in plasma, all drugs reduced the RS ratio relative to pre-treatment control ($P < 0.00001$) (Fig. 4a). However, sterilizing drugs suppressed the RS ratio substantially more than non-sterilizing drugs. Relative to control, rifampin 10 mg kg$^{-1}$, bedaquiline 5 mg kg$^{-1}$, and pyrazinamide reduced the RS ratio 11.1-, 7.6-, and 4.4-fold, respectively. Median RS ratios were 24 (IQR: 24–25), 36 (IQR: 32–40), and 63 (IQR: 59–69), respectively. Non-sterilizing drugs suppressed the RS ratio to a lesser degree. Relative to control, isoniazid, streptomycin, and ethambutol reduced the RS ratio 2.1-, 2.1-, and 2.3-fold, respectively. Median RS ratios were 133 (IQR: 124–140), 133 (IQR: 115–139), and 118 (IQR: 110–129), respectively. RS ratios for sterilizing drugs were significantly lower than for any non-sterilizing drug ($P < 0.00001$ for all comparisons).

We observed dose–response relationships between the drug concentration administered and rRNA synthesis (Fig. 4b). A higher dose of rifampin (30 mg kg$^{-1}$) reduced the RS ratio (median: 16, IQR: 13–18) more than standard-dose rifampin (10 mg kg$^{-1}$, $P = 0.0003$). The standard dose of bedaquiline (25 mg kg$^{-1}$) reduced the RS ratio (median: 10, IQR: 7–12) more than lower-dose bedaquiline (5 mg kg$^{-1}$, $P < 0.00001$). We did not test dose–response relationships for non-sterilizing drugs.

As observed in in vitro experiments, the effects of drugs on rRNA synthesis and bacterial burden were independent (Fig. 4c). Consistent with its negligible bactericidal activity but potent

sterilizing activity, pyrazinamide did not reduce CFU ($P = 0.99$) but did reduce the RS ratio 4.4-fold relative to pre-treatment control. By contrast, streptomycin decreased CFU significantly (0.5 log$_{10}$ CFU lung$^{-1}$, $P = 0.003$) but reduced the RS ratio only 2.1-fold relative to control, consistent with its potent bactericidal activity but low sterilizing activity. Isoniazid and rifampin 10 mg kg$^{-1}$ reduced CFU similarly (1.2 and 1.1 log$_{10}$ CFU lung$^{-1}$, respectively), but rifampin suppressed the RS ratio to a far greater degree ($P < 0.00001$). Bedaquiline had the most potent impact on both the RS ratio and CFU, indicating that it stops rRNA synthesis and results in death. Co-plating revealed minimal acquired drug resistance (Supplementary Fig. 8a).

**A quantitative marker of sterilizing activity in the relapsing mouse model.** Using the conventional high-dose aerosol BALB/c-relapsing mouse model that has historically been the backbone of pre-clinical TB drug and drug regimen evaluation[28], we evaluated the RS ratio as an indicator of sterilizing activity among four regimens with a well-established rank order of sterilizing activity in this model[29–33]. Based on the standard microbiologic relapse outcome, our results confirmed the established rank order of time required for non-relapsing cure, ranging from HRZE (slowest) < PaMZ < BPaL < BPaMZ (fastest) (Fig. 5a and Supplementary Table 2).

After 2, 3, and 4 weeks, the most sterilizing regimen, BPaMZ, was clearly distinguishable from other regimens, suppressing the RS ratio more than the second most potent regimen, BPaL ($P < 0.01$ at each timepoint). In turn, BPaL suppressed the RS ratio more than the third most potent regimen, PaMZ ($P < 0.01$ at each timepoint). The regimens with the lowest sterilizing activity (PaMZ and HRZE) were indistinguishable at weeks 2, 3, and 4 (Fig. 5b).

The decline in the RS ratio tracked with the duration of therapy such that, for each regimen, the longer the duration of treatment, the lower the RS ratio became (trend test $P$ value <0.01 for all regimens) (BPaL shown in Fig. 5c, other regimens in Supplementary Fig. 10). Importantly, the RS ratio was substantially more sensitive than culture. At the end of treatment, most mice were culture negative, but nearly all had quantifiable RS ratios (Fig. 5a),

**a**

| Treatment Weeks | | | | | End of Treatment | | | Post Treatment Weeks | | | Relapse Assessment | | |
|---|---|---|---|---|---|---|---|---|---|---|---|---|---|
| 20 | 16 | 12 | 8 | 4 | Cx pos[a] | RS ratio[b] | RS ratio qf[c] | 4 | 8 | 12 | Relapse[d] | RS ratio[b] | RS ratio qf[c] |
| | | | | BPaMZ | 3/6 | 1.0 | 6/6 | | Off treatment | | 13/15 | 7.9 | 11/15 |
| | | | BPaMZ | | 0/6 | 0.80 | 6/6 | | Off treatment | | 1/15 | 0.46 | 1/15 |
| | | BPaMZ | | | 0/6 | 0.58 | 6/6 | | Off treatment | | 0/15 | nq | 0/15 |
| | | | | BPaL | 6/6 | 4.0 | 6/6 | | Off treatment | | 15/15 | 24.9 | 15/15 |
| | | | BPaL | | 3/6 | 2.2 | 6/6 | | Off treatment | | 9/15 | 0.95 | 12/15 |
| | | BPaL | | | 0/6 | 0.90 | 6/6 | | Off treatment | | 0/15 | 0.30 | 4/15 |
| | | | | PaMZ | 0/6 | 1.6 | 6/6 | | Off treatment | | 3/15 | 0.40 | 9/15 |
| | | | PaMZ | | 0/6 | 0.50 | 6/6 | | Off treatment | | 0/15 | 0.82 | 11/15 |
| | PaMZ | | | | 0/6 | 0.19 | 5/6 | | Off treatment | | 0/15 | 0.78 | 9/15 |
| | | | | 2HRZE/3HR | 4/6 | 1.4 | 6/6 | | Off treatment | | 15/15 | 11.0 | 15/15 |
| | | 2HRZE/3HR | | | 0/6 | 0.77 | 2/6 | | Off treatment | | 12/15 | 4.6 | 12/15 |
| 2HRZE/3HR | | | | | 0/6 | 0.35 | 5/6 | | Off treatment | | 1/14 | 1.2 | 6/14 |

Fig. 5 RS ratio correlated with the sterilizing potency of four regimens in the conventional relapsing mouse model. **a** Summary of treatment durations and outcomes at the end of treatment and the time of relapse assessment in the BALB/c mouse relapse model. At the end of treatment, the numbers of mice with growth of Mtb in culture and with quantifiable RS ratios are shown. The median RS ratio for each treatment is shown. At the time of relapse assessment, the numbers of mice with microbiologic relapse and with quantifiable RS ratios are shown. The median RS ratio for each treatment is shown. **b** RS ratio during treatment weeks 1–4 in lung homogenate from mice treated with BPaMZ (light green), BPaL (dark green), PaMZ, (light brown), or HRZE (dark brown). Circles represent values from individual mice ($N = 3$ per regimen at week 1 and $N = 6$ per regimen at weeks 2, 3, and 4). Solid lines connect median values. **c** RS ratio results following 4, 8, and 12 weeks of treatment with BPaL. Circles represent individual mice ($N = 6$ per regimen and duration) at end of treatment (EOT). Squares represent mice at the time of relapse determination ($N = 15$ per regimen and duration) after a 12-week drug holiday. **d** CFU per lung at the same timepoints in the same mice as in **b** with the same number of mice represented in the same color scheme.

indicating that the RS ratio can quantify drug effect beyond the point at which mice become culture negative.

The RS ratio tracked with the propensity to relapse. For example, after 4 weeks of treatment with BPaL, the RS ratio was partially suppressed (median = 4.0) but, after a 12-week drug holiday, the RS ratio rebounded (median = 24.9) (Fig. 5d) and all mice relapsed. An additional 4 weeks of BPaL suppressed the RS ratio further (median = 2.2) at the end of treatment, but, after a 12-week drug holiday, the RS ratio remained quantifiable in 12 of 15 (80%) mice and 9 of 15 (60%) mice had microbiologic relapse. The longest BPaL arm (12 weeks) suppressed the end of treatment RS ratio to the lowest level (median = 0.95). After a 12-week drug holiday, the RS ratio was quantifiable in only four mice and no mice relapsed. A similar association between treatment duration, suppression of the RS ratio, and non-relapsing cure was observed for all regimens (Supplementary Fig. 10). An additional smaller study that included only HRZE and BPaL identified the same results (Supplementary Fig. 11).

Comparison of RS ratio and CFU results reinforces that they measure orthogonal properties. Consistent with in vitro testing of single drugs, combination regimens rapidly inhibited rRNA synthesis (10- to 100-fold decrease in RS ratio during week 1), well before there was a meaningful decline in CFU burden (Fig. 5b, c). During weeks 1–4, CFU grouped PaMZ with BPaL while the RS ratio grouped PaMZ with HRZE. However, after 20 weeks, PaMZ suppressed the RS ratio significantly more ($P = 0.03$) than HRZE. PaMZ was also more effective than HRZE in preventing both rebound in the RS ratio and relapse following the completion of treatment. After 14 weeks of HRZE and a 12-week drug holiday, the RS ratio rebounded (median = 11.0) and all mice relapsed. By contrast, after 12 weeks of PaMZ and a 12-week drug holiday, the RS ratio remained suppressed (median = 0.40) and 20% of mice relapsed.

**Effect of treatment on the RS ratio in human TB**. To begin evaluation of the RS ratio as a marker of treatment response in humans, we quantified the RS ratio in serial sputa from 17 Ugandan and 28 Vietnamese patients treated with HRZE for drug-susceptible pulmonary TB (Fig. 6a). In these geographically distinct populations with different dominating circulating Mtb lineages, HRZE rapidly suppressed the RS ratio. In Uganda, the

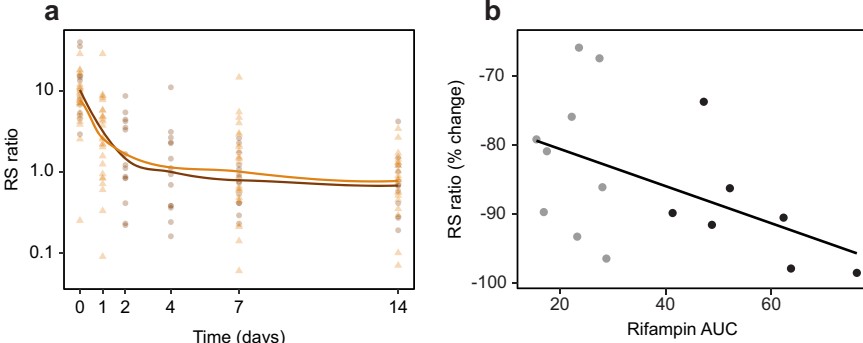

**Fig. 6 Sputum RS ratio declines rapidly with standard treatment and a dose-response relationship is evident. a** Longitudinal change in the RS ratio in serial sputa among Ugandan and Vietnamese adults treated with HRZE for drug-susceptible TB. Brown and orange dots indicate individual patients from Uganda and Vietnam, respectively. The solid line is a Lowess fit. **b** Association between rifampin exposure (AUC) and change in sputum RS ratio between pre-treatment and day 14 among Beninese patients treated with HRZE with rifampin 10 mg kg$^{-1}$ (gray) or 15 mg kg$^{-1}$ (black).

median RS ratio declined 6.3-fold (IQR: 3.4–17.3-fold) from baseline after 2 days and 13.7-fold (IQR: 8.1–20.7-fold) after 7 days. In Vietnam, the median RS ratio declined 3.0-fold (IQR: 1.5–5.1-fold) after 1 day and 9.3-fold (IQR: 3.8–21.7-fold) after 7 days. To establish a dose–response relationship between rifampin exposure and the RS ratio, we collected serial sputa and serum pharmacokinetic data from 19 Beninese patients treated for drug-susceptible pulmonary TB with HRZE including standard 10 mg kg$^{-1}$ rifampin or 15 mg kg$^{-1}$ rifampin. Higher rifampin area under the curve (AUC) was associated with a greater decrease in the RS ratio over 14 days (Pearson correlation coefficient = −0.50, $P = 0.01$) (Fig. 6b).

## Discussion

We discovered that drugs and drug regimens that shorten the duration of TB treatment inhibit *Mtb* rRNA synthesis more than less potent drugs and regimens. The time needed to cure TB is determined primarily by drug activity against residual *Mtb* populations that survive initial drug exposure[2,4,10,11]. This activity has not previously been directly measurable. By quantifying the impact of drugs on rRNA synthesis rather than enumerating bacterial burden, the RS ratio provides a practical metric of drug activity that may enhance pharmacodynamic monitoring and accelerate development of shorter TB treatment regimens.

Historically, the treatment-shortening activity of drugs has been characterized by observing the effects of a drug in a series of animal relapse studies and human clinical trials[2]. The length and expense of human and animal trials has impeded evaluation of candidate regimens. Several features suggest that the RS ratio may accelerate evaluation of drugs and regimens. First, the RS ratio measures a property that is distinct from bacterial burden. Drugs and regimens frequently affect CFU and the RS ratio differently. For example, bedaquiline suppressed the RS ratio within hours in vitro, indicating near-cessation of rRNA synthesis, yet CFU did not decline appreciably for 8–12 days. Similarly, the potent sterilizing agent pyrazinamide suppressed the RS ratio in mice but had no effect on CFU. Conversely, the effect of PaMZ on CFU early in treatment was greater than its effect on the RS ratio. The RS ratio is not a proxy for bacterial burden. Second, in both mice and humans, there was a dose-response in which higher doses of sterilizing drugs suppressed the RS ratio to a greater degree than lower doses. Finally, the RS ratio correlated with regimen sterilizing activity in the conventional relapsing mouse model. The regimens that cured TB fastest were those that suppressed the RS ratio more rapidly and most profoundly. Collectively, these

findings suggest that the RS ratio may provide a needed practical marker of the sterilizing activity of drugs[10,13].

The RS ratio was able to quantify drug effect beyond the point at which all mice were culture negative. Understanding of sterilizing activity has long been hamstrung by the limited sensitivity of culture. Like humans, mice become culture negative before they are cured[28,34]. This is highlighted by our results with HRZE. After 16 weeks of HRZE, all mice were culture negative. Yet, when held for a 12-week drug holiday, 80% of companion mice had microbiologic relapse. By providing sensitive, precise quantitative information on drug effect through the entire course of treatment, the RS ratio opens a window on the critical but hitherto inaccessible late sterilizing phase. This has immediate practical implications for regimen evaluation in pre-clinical animal models. Because it is measured in a relatively small number of animals (3–6 mice) early in treatment, the RS ratio saves time, resources, and animals. By greatly increasing the speed with which large numbers of drug regimens can be ranked in pre-clinical models, the RS ratio should accelerate selection of regimens for human testing.

The central challenge to shortening TB treatment is eliminating the drug-tolerant persister population that withstands the initial rapid killing phase. Genetically drug-susceptible bacterial populations that have survived prolonged drug exposure (such as those studied here) have been defined functionally as drug-tolerant persisters[35]. The physiologic state of persisters is uncertain[35], with evidence supporting the existence of both replicating[36] and non-replicating phenotypes[37,38]. The concept that certain persisters may continue replication is based on reports that *M. smegmatis* sustains ongoing replication during lethal isoniazid exposure[36] while sterilizing drugs, bedaquiline and rifampin, halt *Mtb* replication[39]. Since it is well-established that rRNA synthesis and bacterial replication are fundamentally coupled[16,17], our results with the RS ratio seem consistent with drug-specific effects on *Mtb* replication. Conversely, our findings are not necessarily at odds with the conventional model that persisters are non-replicating with a low basal level of transcriptional, translational, and metabolic activity[37,38].

This work focuses attention on the down-stream physiologic consequences of drug stress rather than the specific drug mechanism of action. For all but rifampin, the connection between mechanism of action and suppression of rRNA synthesis is indirect. The interaction of a drug with its target molecule initiates a cascade of indirect secondary damage, perturbing other cellular processes. Secondary effects are complex and currently difficult to predict based on drug mechanism alone.

We see several non-exclusive possibilities for how inhibition of rRNA synthesis may accelerate time to cure. First, when drug stress rapidly and profoundly impairs the ability of a bacterium to synthesize a key macromolecule (rRNA), this may be a signal of injury that is incompatible with pathogenicity or long-term viability. A bacterium that cannot synthesize rRNA is likely unable to remodel its proteome, committing it to a single physiologic program and limiting its fitness to respond dynamically to its environment. A depleted, incapacitated *Mtb* population may be less capable of withstanding immune stress or may even elicit a different immune response. Finally, a bacterium that cannot synthesize rRNA will be unable to replicate[16,17]. A drug or regimen that abrogates rRNA synthesis will halt the production of new bacilli during treatment.

Our results suggest several future studies. First, understanding the full range of possible drug effects on rRNA synthesis will require testing emerging new chemical entities with additional mechanisms of action. A limitation of this report is that we did not test drugs that may have sterilizing activity, including moxifloxacin, clofazimine, and linezolid, individually in vitro and in mice. Second, we cannot be certain that the association between RS ratio and treatment shortening activity is generalizable to non-rifamycin, non-bedaquiline-based regimens that we have not yet tested. It remains possible that inhibition of RNA or ATP synthesis suppresses the RS ratio in a way that is independent of sterilizing activity. Confirming the RS ratio as a practical surrogate for relapse in animals will require additional relapse studies with diverse regimens in multiple animal models. Finally, while this report provides proof-of-concept data in humans, the value of the sputum RS ratio remains to be determined. Ongoing clinical trials (ClinicalTrials.gov Identifier: NCT02410772) are testing the RS ratio as pharmacodynamic monitoring tool in humans.

In summary, this study has identified a key difference in how sterilizing and non-sterilizing drugs and regimens affect *Mtb* rRNA synthesis. The RS ratio provides a needed molecular metric of drug activity that is based a key physiologic property rather than recapitulation of bacterial burden. The RS ratio may enable more intelligent design and evaluation of candidate regimens, accelerating development of regimens that can cure TB faster.

## Methods

**In vitro oxygen depletion model**. *M. tuberculosis* H37Rv was grown in 125 ml Erlenmeyer flasks in 50 ml DTA medium (Dubos broth (BD Difco) supplemented with 0.5% bovine serum albumin (Research Products International), 0.05% Tween 80, and 0.75% glucose, pH 6.6) stirring at 200 r.p.m. with $50 \times 8$ mm stir bars using a Micro-Stir magnetic stirrer (Wheaton) at 37 °C until mid-log ($OD_{600}$ 0.4). Cultures were diluted to $OD_{600}$ 0.004 in DTA medium in 16 ml volumes in sterile glass $20 \times 125$ mm tubes. Stopcock grease was applied to the threads of the glass tubes and tubes were sealed with phenolic caps. These cultures were stirred at 200 r.p.m. with $12 \times 4.5$ mm stir bars using a rotary magnetic tumble stirrer (V&P Scientific) for rapid oxygen depletion[18]. Cell pellets to assess rRNA synthesis ratios were collected as detailed in Supplementary Information every 12 h starting at day 4, after cultures had begun to become hypoxic. Growth rates were determined based on optical density readings at 600 nm ($OD_{600}$) every 6 h for the duration of the experiment.

**Antibiotic killing assay**. *M. tuberculosis* Erdman was grown in 125 ml Erlenmeyer flasks in 50 ml 7H9 medium (7H9 broth (BD Difco) supplemented with 0.2% glycerol, 0.5% bovine serum albumin (Research Products International), 0.05% Tween 80, 0.2% glucose, and 0.085% sodium chloride, pH 6.6) stirring at 200 r.p.m. with $50 \times 8$ mm stir bars using a Micro-Stir magnetic stirrer (Wheaton) at 37 °C with 5% $CO_2$ until mid-log ($OD_{600}$ 0.4). Cultures were diluted to $OD_{600}$ 0.05 in 7H9 medium in 5 ml volumes in sterile glass $20 \times 125$ mm tubes and stirred at 200 r.p.m. for 18 h at 37 °C with $CO_2$ and $12 \times 4.5$ mm stir bars using a rotary magnetic tumble stirrer (V&P Scientific). RNA was extracted from cell pellets as described in the Supplementary Information.

Antibiotics were added as follows: rifampin 1 μg ml$^{-1}$ (Chem-Impex International), isoniazid 0.5 μg ml$^{-1}$, streptomycin 5 μg ml$^{-1}$ (Fisher), ethambutol 4 μg ml$^{-1}$ (MP Biomedicals), and bedaquiline 8 μg ml$^{-1}$ (NIH AIDS Reagent Program). Cultures were plated for enumeration on DTA agar supplemented with

0.4% of activated charcoal and co-plated on 7H10 agar (7H10 agar base (BD Difco) supplemented with 0.005% oleic acid, 0.5% bovine serum albumin, 0.2% glucose, 0.085% sodium chloride, and 0.0004% catalase) for resistance determination. Antibiotic concentrations in plates were as follows: rifampin 8 μg ml$^{-1}$, isoniazid 0.625 μg ml$^{-1}$, streptomycin 8 μg ml$^{-1}$, ethambutol 8 μg ml$^{-1}$, and bedaquiline 1 μg ml$^{-1}$.

**Quantification of the rRNA synthesis ratio via droplet digital PCR**. RNA was reverse transcribed with SuperScript VILO cDNA synthesis kit (Invitrogen) according to the manufacturer's protocol, except that reverse transcription at 42 °C was performed for 120 min. Transcript copies were quantified using the QX100 Droplet Digital PCR system (Bio-Rad). Primers and probe sequences and information are in Supplementary Table 3. Reaction were run in duplex, ETS1 with 23S and ITS1 with 23S with ddPCR SuperMix for Probes (no dUTP) (Bio-Rad). All primers were 900 nM final concentration and all probes were 250 nM final concentration. The thermocycling conditions for all ddPCR reactions were: initial denaturation at 95 °C for 10 min, 40 cycles of 94 °C for 30 s and 60 °C for 60 s with a 2 °C s$^{-1}$ ramp rate, and a final hold at 98 °C for 10 min. The ratio of ETS1/23S and ITS1/23S was calculated within each duplexed reaction by QX100 Droplet Digital PCR system software (Bio-Rad).

**RNA sequencing**. RNA extracted from in vitro samples was reverse transcribed and prepared for sequencing using the Truseq Stranded Total RNA kit (Illumina), omitting the ribosomal depletion step but otherwise following the manufacturer's protocol. cDNA libraries were sequenced on a NovaSEQ 6000 (Illumina) at the University of Colorado, Anschutz Medical Campus Genomics and Sequencing Core. Sequence quality was evaluated using FastQC and adapters were trimmed using BBDuk (https://sourceforge.net/projects/bbmap/) with kmer = 23 and mink = 11. High-quality sequences were randomly subsampled to 100,000 sequences per sample with BBTools (v 35.85)[40] (https://sourceforge.net/projects/bbmap/) and mapped to *M. tuberculosis* Erdman ATCC35801 (accession number NC_020559.1) using Bowtie2 (ref. [41]) with the default parameters, followed by visualization with IGV[42]. Bioinformatics analysis was performed on the Colorado Center for Personalized Medicine High Performance Computing Center at the University of Colorado, Anschutz Medical Campus.

**Animal efficacy studies**. All animal studies were performed at Colorado State University in a certified animal biosafety level III facility. Ethics oversight was provided by the Colorado State University Animal Care and Use Program which is PHS Assured (A3572-01), USDA Registered (84-R-0003), and AAALAC accredited (#000834). The IACUC approved CSU protocol number is 17-7701A.

Mice were housed socially (2–5 animals per cage) in a certified ABSL-3 facility in HEPA filter equipped techniplast cages on autoclaved bedding changed every 7–14 days. Mice had access to irradiated chow and water ad libitum. Twelve-hour light/dark cycles were employed and mice were maintained at temperatures between 65 and 75 °F with 40–60% humidity.

**Infection of mice**. Aerosol infection of mice with *M. tuberculosis* Erdman employed a Glas-Col inhalation exposure system[43,44].

**Drug delivery and dose**. Isoniazid, pyrazinamide, linezolid, and ethambutol were administered by oral gavage in a 0.2 ml volume at 25, 150, 50, and 100 mg kg$^{-1}$, respectively. Rifampin, bedaquiline, and pretomanid were administered by oral gavage in 0.2 ml volume at 10 or 30, 5 or 25, and 50 or 100 mg kg$^{-1}$, as indicated for each study. Streptomycin was given by subcutaneous injection at 200 mg kg$^{-1}$ in 0.2 ml volume. In cases where individual mice were administered two oral drugs, each individual oral dose was separated by 1 h. The global standard HRZE regimen was prepared by combining isoniazid, ethambutol, and pyrazinamide in 0.2 ml volume and giving by oral gavage 1 h following delivery of rifampin as a separate oral dose. The PaMZ regimen was prepared by combining moxifloxacin and pyrazinamide and delivering ~4 h after delivery of pretomanid as a separate oral dose. The BPaL regimen was prepared by combining bedaquiline and pretomanid and delivering ~4 h prior to the delivery of linezolid as a separate oral dose. The BPaMZ regimen was prepared by combining moxifloxacin and pyrazinamide and delivering ~4 h after delivery of bedaquiline and pretomanid both given as separate oral doses ~1 h apart. All treatments were given once daily, 5 days a week (Monday through Friday).

**C3HeB/FeJ mouse experiments**. Six- to 8-week old C3HeB/FeJ (Jackson Laboratories) female mice were exposed to a low-dose aerosol of *M. tuberculosis* Erdman using $1.5 \times 10^6$ CFU ml$^{-1}$ to achieve ~50–75 CFU in the lungs of each mouse[23,24]. Treatment was initiated on day 71 at the time when necrotic lesions have fully developed[23] and continued for 4 weeks. Each mouse was individually euthanized by $CO_2$ narcosis followed by cardiac puncture. Lung lobes were photographed and Type I caseating necrotic lesions of <5 mm were excised and placed into phosphate-buffered saline with 4% paraformaldehyde at 4 °C for 3 days prior to further processing.

**BALB/c mouse chronic TB model**. Six- to 8-week-old female pathogen-free BALB/c mice (Charles River Laboratories) were exposed to a low-dose aerosol of *M. tuberculosis* Erdman-Lux[44] using $2 \times 10^6$ CFU ml$^{-1}$ to achieve ~71 CFU in the lungs of each mouse. Three mice were individually euthanized by $CO_2$ narcosis on day 1, day 7, day 25, and day 53 post aerosol infection. Bacterial lung burdens were determined from the left lung lobe. Upper right lung lobes (superior and middle lobes) were flash frozen in liquid nitrogen prior to RNA extraction as described in the Supplementary Information.

**BALB/c mouse high-dose aerosol infection model**. Six to 8-week-old female pathogen-free BALB/c mice (Jackson Laboratories) were exposed to high-dose aerosol of *M. tuberculosis* Erdman from broth culture (OD$_{600}$ ~0.8) to achieve deposition of ~3.8 log10 CFU in the lungs of each mouse[45,46]. Treatment was initiated on day 11 post aerosol and continued for 4 weeks. Groups of six mice were individually euthanized by $CO_2$ narcosis on day 11, prior to treatment initiation, and on the last day of treatment, to determine the bacterial loads in lungs. The left and lower right lung lobes (inferior and post-caval lobes) were used for bacterial enumeration. Upper right lung lobes (superior and middle lobes) were flash frozen in liquid nitrogen prior to RNA extraction as detailed in the Supplementary Information.

**BALB/c mouse relapse model**. Six- to 8-week-old female pathogen-free BALB/c mice (Jackson Laboratories) were exposed to high-dose aerosol of *M. tuberculosis* Erdman from broth culture (OD$_{600}$ ~0.8) to achieve deposition of ~4.3 log$_{10}$ CFU in the lungs of each mouse[45,46]. Treatment was initiated on day 11 post aerosol and continued for up to 20 weeks. Groups of 3–6 mice, as indicated, were individually euthanized by $CO_2$ narcosis on day 11, prior to treatment initiation, and one day following the last day of treatment, to determine the bacterial loads in the lungs. Additional groups of 15 mice each from each treatment group were placed on a 12-week drug holiday to allow bacterial relapse[25,30–32,45]. The left and lower right lung lobes (inferior and post-caval lobes) were used for bacterial enumeration. Upper right lung lobes (superior and middle lobes) were flash frozen in liquid nitrogen prior to bead beating and RNA extraction as described in the Supplementary Information.

**Enumeration of CFU from lungs**. The number of viable organisms was determined by serial dilutions of homogenates (Precellys Evolution, Bertin) prepared in phosphate-buffered saline plus 10% (w/v) bovine serum albumin from whole lungs (C3HeB/FeJ mice) or indicated lung lobes (BALB/c mice) and plating on 7H11-OADC agar plates containing 0.4% (w/v) activated charcoal to prevent drug carry-over. Colonies were enumerated after at least 21 days of incubation at 37 °C. For relapse assessments, tissues were homogenized in phosphate-buffered saline and plated in their entirety on 7H11-OADC agar plates without activated charcoal.

**Single-bacillary ISH**. Mouse lung was formaldehyde-fixed and paraffin-embedded and stained by using multiplexed-ISH kit (Advanced Cell Diagnostics) according to the manufacturer's instructions[47]. Specimens were directly placed into 4% PFA upon excision and fixed for 48 h at 4 °C before embedding. Next, 2 μm tissue sections were cut from FFPE blocks and mounted onto Superfrost Plus microscope slides (Fisher Scientific) prior to use. Multiplex-ISH was visualized after labeling with fluorescein isothiocyanate and cyanine 3.5 ([47]). Whole-slide digital images were acquired at ×40 magnification using the Axio Scan.Z1 slide scanning fluorescence microscope (Zeiss) or at ×63 magnification using the SP8 laser scanning confocal system (Leica). Image analysis was performed using the ilastik machine-learning-based (bio)image analysis (www.ilastik.org)[48]. The ilastik plugin for ImageJ (v. 2.1.0/1.53c) was used to export data from each region of interest in the ilastik HDF5 format.

**Image analysis statistics**. The data tables were exported from ilastik as "*.csv" files for analysis using R software (version 3.5.2, R Foundation for Statistical Computing, www.R-project.org). Background intensity was corrected for each channel by subtracting the minimum 30-pixel neighborhood intensity from the MFI for each object. After correcting for background, the MFIs were analyzed and reported in log$_{10}$. The heterogeneity of signal intensities within the inflammatory rim and the caseum was calculated by the variance. MFI values of each channel based on location were compared using an *F*-test. Measured MFIs in the two channels were tested for within group variance in both the inflammatory rim and the caseum. A non-parametric Kruskal–Wallis one-way analysis of variance was used to determine dominance in median values in observed differences between ratio values, channels, and locations.

**Human study subjects**. This manuscript includes three human studies. The first was a longitudinal study of TB patients treated under routine care, conducted across eight outpatient clinics in Hanoi, Vietnam, by the US CDC TB Trials Consortium at the UCSF/Vietnam National TB Programme network, entitled "Study 36: A Platform for Assessment of TB Treatment Outcomes An Observational Study of Individuals Treated for Pulmonary Tuberculosis." The second was a longitudinal observational study in Uganda that included 17 adult inpatients (male

and female) treated for drug-susceptible TB per local guidelines with the global standard four-drug regimen at standard doses. An analysis of *Mtb* mRNA in sputum from this cohort has been published[49]. The third was a biomarker substudy embedded in the Benin site of the RAFA trial which enrolled adults living with HIV who were co-infected with drug-susceptible TB. Patients were randomized to either a control arm, which was standard of care at the time (standard antitubercular treatment with 10 mg kg$^{-1}$ doses of rifampicin and start of ART 8 weeks thereafter), or early start of ART (2 weeks after initiating antituberculosis treatment), or receive a high dose of rifampicin in the first 8 weeks of TB treatment (50% dose increase, i.e., 15 kg$^{-1}$, and start of ART 8 weeks after initiating antituberculosis treatment). Aspects of the RAFA trial have been published[50]. All participants in the three cohorts (Vietnam, Uganda, and Benin) provided written informed consent for the use of their sputa and clinical information for the purpose of developing novel biomarkers of treatment response. Ethical approval and supervising institutional review boards are fully described in Supplementary Information.

**Pharmacokinetic sampling in the RAFA trial**. Four weeks after initiating treatment, patients were admitted overnight before pharmacokinetic sampling. Five serial blood samples were drawn: pre-dose (~15 min before a dose) and 2-, 3-, 6-, and 10-h post-dose. Blood samples were processed, and plasma was stored immediately at −80 °C before transfer on ice to the analytical laboratory (Division of Clinical Pharmacology, University of Cape Town, South Africa). Plasma samples were analyzed using a validated liquid chromatography-tandem mass spectroscopy (LC-MS/MS) assay[50].

**Pharmacokinetic/pharmacodynamic modeling**. The population pharmacokinetic model for rifampin was developed using nonlinear mixed-effects modeling in software NONMEM (v 7.3; Icon Development Solutions). Absorption of rifampin was described using a first-order absorption model, with a delay, using a chain of transit compartments[51]. One-compartment disposition model was used to describe pharmacokinetics distribution of rifampin[52]. Allometric scaling was applied to all clearance and volume of distribution parameters to account for the effect of body size using total body weight (TBW), fat-free mass (FFM), or body fat[53]. Between-patient variability in PK parameters (clearance, volume, absorption rate constants) was implemented using a log normal distribution. The final pharmacokinetic model was validated using internal validation techniques, such as visual predictive check and non-parametric bootstrapping. Estimates of individual areas under the concentration–time curve (AUC) and maximum serum concentrations ($C_{MAX}$) were derived from the models using integration from the system of ODE equations and individually estimated primary pharmacokinetic parameters (CL, V, $k_a$).

The PKPD model was built using a sequential approach. The longitudinal pharmacodynamic biomarker rRNA synthesis ratio was modeled using a linear model where baseline and rate of change were estimated (intercept-slope) model from the data. Between-subject variability was implemented on baseline and slope parameters. Full variance–covariance model was estimated for baseline–slope parameter distribution. After a baseline model was developed which describes on treatment biomarker response as a function of treatment and baseline, we then evaluated rifampin pharmacokinetics for its significance as a covariate impacting the slope (rate of change), using a linear model. Therefore, the change from baseline in rRNA synthesis ratio was described in a final model as a function of, treatment, baseline and drug pharmacokinetics predictors (AUC or $C_{MAX}$). The likelihood ratio test, which compares −2 log likelihood between two nested models, was used to assess significance. A baseline model was developed first, followed by the addition of AUC or $C_{MAX}$.

**Statistics and reproducibility**. The decision on whether a parametric or non-parametric test should be used was based on the Shapiro–Wilk test. All statistical tests were two-tailed unless otherwise noted. For single exposure in vitro, data were evaluated by a Kruskal–Wallis one-way analysis of variance followed by a planned multiple comparison analysis. For murine studies, data were evaluated by a one-way analysis of variance followed by a multiple comparison analysis of variance by a one-way Tukey test. The association between rRNA synthesis ratio and rifampin PK was done using simple linear regression. For statistical analyses performed on MFI values, the one-way Kolmogorov–Smirnov test of significance comparing the empirical CDFs was first performed, followed by the Kruskal–Wallis test. Differences were considered significant at the 95% level of confidence ($P < 0.05$). SigmaPlot software (v 11) and R (v 3.2.3) were used for data manipulation, plotting, and post-modeling analysis.

**Reporting summary**. Further information on research design is available in the Nature Research Reporting Summary linked to this article.

## Data availability
All raw sequencing data have been deposited in the Sequence Read Archive (SRA) under BioProject accession PRJNA615137. Individual samples have the following BioSample accession numbers: untreated, SAMN14446914; rifampin, SAMN14446915; isoniazid, SAMN14446916; streptomycin, SAMN14446917; ethambutol, SAMN14446918;

bedaquiline, SAMN14446919. Data files related to image analysis of C3HeB/FeJ mouse lesions are available at https://github.com/JoshuaVasquezLab/Walter-et-al.2021 (ref. [54]). Other source data are provided with this paper in the Source Data file.

## Code availability

Code related to image analysis of C3HeB/FeJ mouse lesions can be found at https://github.com/JoshuaVasquezLab/Walter-et-al.2021 (ref. [54]).

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

## Acknowledgements

This N.D.W., R.S., J.L.D., P.N., and G.S. acknowledge funding from the US National Institutes of Health (1R01AI127300-01A1). N.D.W. and M.I.V. acknowledge funding from the US National Institutes of Health (1R21AI135652-01) and the University of Colorado Department of Medicine Team Science Award. N.D.W., M.I.V., G.T.R., P.N., and R.S. acknowledge funding from the Bill & Melinda Gates Foundation (OPP1213947). N.D.W., M.I.V., and G.T.R. acknowledge funding from the SPARK program at the University of Colorado Anschutz Medical Campus. N.D.W. and C.W. acknowledge funding from the Colorado Clinical and Translational Sciences Institute Novel Clinical and Translational Methods Pilot Program. N.D.W. acknowledges funding from Veterans Affairs (1IK2CX000914-01A1 and 1I01BX004527-01A1) and from the Doris Duke Charitable Foundation Clinical Scientist Development Award. J.J.V. acknowledges funding from the US National Institutes of Health (K01HL140804, R21 AI116295, P30-AI027763) and the Robert Wood Johnson Foundation Harold Amos Medical Faculty Development Program and the UCSF Nina Ireland Program for Lung Health. R.S acknowledges funding from the US National Institutes of Health (R01AI135124). P.N. acknowledges funding from the US National Institutes of Health (5R01AI127300). J.L.D. acknowledges funding from the US National Institutes of Health (R21 AI101714). E.M. acknowledges funding from the US National Institutes of Health (D43 TW009607). L.H. acknowledges funding from the US National Institutes of Health (R01 HL090335, R01 HL128156, R01 HL143998, K24 HL087713). C.S.M. acknowledges funding from the European & Developing Countries Clinical Trials Partnership (PACTR201105000291300). H.M. acknowledges funding from the Wellcome Trust (206379/Z/17/Z). Data collection and sharing for this project was additionally supported by the Randall MacIver Trust at the University of Colorado. Tissue processing, imaging, and analysis was supported by the technology and staff of the following core labs: UCSF Anatomic Pathology at ZSFG, Biological Imaging Development CoLab (BIDC) at UCSF Parnassus Heights, Histology & Biomarker Core of the UCSF Helen Diller Family Comprehensive Cancer Center, the CSU Experimental Pathology Core Facility and the CSU Molecular Quantification Core Facility. We gratefully acknowledge the assistance of John Anderson and Dr. Joseph Russo for training and implementation of ddPCR.

## Author contributions

N.D.W., M.I.V., S.E.M.B., G.T.R., M.R., A.J.L., G.S., G.D., J.L.D., K.M., C.B., P.N., R.S., and J.J.V. conceived of the project and planned the experiments. N.D.W., S.E.M.B., G.T.R., and M.I.V. drafted the manuscript. S.E.M.B., M.R., and M.I.V. generated and analyzed in vitro data. V.A.E., B.K.P., M.E.R., A.A.B., and G.T.R. generated and analyzed murine data. E.M., L.H., and W.W. conducted human studies in Uganda. N.D.S., B.D.J., C.S.M., and D.A. conducted human studies in Benin. N.V.N., H.V.N., A.T.V.N., and H.P. conducted human studies in Vietnam. E.M., F.E.-W., B.A.-R., D.K., and J.J.V. generated and analyzed in situ hybridization data. C.D.-A., K.R., S.C.B., C.W., V.O. and B.G. generated and analyzed molecular data. M.G.-C., H.M. and R.S. conducted PKPD analyses.

## Competing interests

The University of Colorado has filed a patent application entitled "Methods of Evaluating Treatment Efficacy and/or Treatment Duration In Mycobacterial Diseases [Application 16/632,310, filed January 17, 2020, currently pending] pertaining to use of pre-rRNA ratios for monitoring treatment efficacy. Inventors are N.D.W., M.I.V., G.T.R., A.J.L., G. D., G.S., P.N., and J.L.D. The remaining authors declare no competing interests.

## Additional information

[1]Rocky Mountain Regional VA Medical Center, Aurora, CO, USA. [2]Division of Pulmonary Sciences and Critical Care Medicine, University of Colorado Anschutz Medical Campus, Aurora, CO, USA. [3]Consortium for Applied Microbial Metrics, Aurora, CO, USA. [4]Department of Immunology and Microbiology, University of Colorado Anschutz Medical Campus, Aurora, CO, USA. [5]Mycobacteria Research Laboratories, Department of Microbiology, Immunology, and Pathology, Colorado State University, Fort Collins, CO, USA. [6]Department of Epidemiology, Colorado School of Public Health, Aurora, CO, USA. [7]Division of Infectious Diseases and Geographic Medicine, Stanford University, Palo Alto, CA, USA. [8]Integrated Center for Genes, Environment, and Health, National Jewish Health, Denver, CO, USA. [9]Computational Bioscience Program, University of Colorado Anschutz Medical Campus, Aurora, CO, USA. [10]Infectious Disease Research Collaboration, Kampala, Uganda. [11]Department of Biochemistry, Makerere University, Kampala, Uganda. [12]Division of Pulmonary and Critical Care Medicine, University of California San Francisco, San Francisco, CA, USA. [13]Division of HIV, Infectious Diseases and Global Medicine, University of California San Francisco, San Francisco, CA, USA. [14]Zuckerberg San Francisco General Hospital, San Francisco, CA, USA. [15]Department of Epidemiology of Microbial Diseases, Yale School of Public Health, New Haven, CT, USA. [16]Pulmonary, Critical Care, and Sleep Medicine Section, Yale School of Medicine, New Haven, CT, USA. [17]Vietnam National TB Programme/UCSF Research Collaboration Unit, Hanoi, Vietnam. [18]Laboratoire de Référence des Mycobactéries, Cotonou, Benin. [19]Mycobacteriology Unit, Institute of Tropical Medicine, Antwerp, Belgium. [20]London School of Hygiene and Tropical Medicine, London, UK. [21]UNICEF/UNDP/World Bank/WHO Special Programme on Research and Training in Tropical Disease, Geneva CH, Switzerland. [22]Division of Clinical Pharmacology, University of Cape Town, Cape Town, South Africa. [23]Wellcome Centre for Infectious Diseases Research in Africa, Institute of Infectious Disease and Molecular Medicine, University of Cape Town, Cape Town, South Africa. [24]Department of Bioengineering and Therapeutic Sciences, University of California San Francisco, San Francisco, CA, USA. [25]Division of Experimental Medicine, University of California San Francisco, San Francisco, CA, USA. [26]Bill & Melinda Gates Medical Research Institute, Cambridge, MA, USA. [27]Bill and Melinda Gates Foundation, Seattle, WA, USA. [28]UCSF Center for Tuberculosis, University of California, San Francisco, CA, USA. [29]These authors contributed equally: Nicholas D. Walter, Sarah E. M. Born, Gregory T. Robertson. ✉email: nicholas.walter@cuanschutz.edu; martin.voskuil@cuanschutz.edu

