## [Peer Review File · Nature Communications]

REVIEWERS' COMMENTS

Reviewer #2 (Remarks to the Author):

I commend the authors on their extensive and rigorous responses to my (and the other reviewers') concerns. The additional mouse relapse experiments strongly support the conclusions and proposed implementation of the rRNA synthesis ratio as readout of drug effect and marker of sterilizing activity. They have provided extensive additional data and discussion, further strengthening the manuscript. This is an important and beautifully conducted study.

Reviewer #3 (Remarks to the Author):

This is a revision to the article that I previously reviewed. The authors have provided detailed responses to every reviewer comment (in fact, I've never seen author responses that were so thoughtful, and complete). They enhanced the original work by conducting large additional animal studies (funded by Bill & Melinda Gates Foundation-- authors added) and by adding patient data from a second continent (Vietnam, added to existing data from Uganda). They answered my questions in full, as well as those of fellow reviewers. I have no further concerns.

Reviewer #4 (Remarks to the Author):

The concept of using pre-rRNA/rRNA ratios as a surrogate for bacterial killing in murine models is innovative, novel, and of interest. The authors have also supplemented their data with a nice new experiment looking at RS in murine relapse across 4 regimens and 3 treatment durations.

1. Critique on RS as an indicator of sterilization vs transcriptional blockade or ATP depletion. "It is not at all surprising that a drug that inhibits RNA polymerase (RIF) and a drug that dramatically depletes the key RNA precursor ATP (BDQ) would slow RNA biosynthesis in *M.tb* and lead to low rRNA synthesis ratios by virtue of the unstable ETS1 species being cleared rapidly while the stable 23S rRNA lingering. So the finding that RIF and BDQ reduce rRNA synthesis ratios may be coincidental to the drugs' mechanisms of action and have nothing to do with their sterilizing activities"

The authors make the point that their relapse study favors the argument that RS is measuring sterilizing ability, this point is still not proven. It would be helpful to readers to point out alternative interpretations in the discussion

2. Critique on why not test other known sterilizing drugs. Other known sterilizing drugs which do not have mechanisms of action that inhibit RNA synthesis were not tested in the in vitro analysis (Fig. 3). These include pyrazinamide, moxifloxacin, linezolid, and clofazimine.

Disappointingly, the authors make rhetorical arguments here to defend the omission of testing these monotherapies. Their point about PZA is credible, but to make their results as convincing as possible, the others should have been tested. And having tested these other drugs which do not act by the same mechanisms as RIF and BDQ is pertinent to point #1 above. It would be helpful to readers to point out that the RS method was not tested with sterilizing monotherapies such as moxifloxacin, linezolid, and clofazimine somewhere in the discussion.

Response to Reviewers

We thank the reviewers for their additional consideration of this manuscript.

Reviewer 2

I commend the authors on their extensive and rigorous responses to my (and the other reviewers') concerns. The additional mouse relapse experiments strongly support the conclusions and proposed implementation of the rRNA synthesis ratio as readout of drug effect and marker of sterilizing activity. They have provided extensive additional data and discussion, further strengthening the manuscript. This is an important and beautifully conducted study.

We are grateful for the reviewers' critiques that led to these improvements.

Reviewer 3

This is a revision to the article that I previously reviewed. The authors have provided detailed responses to every reviewer comment (in fact, I've never seen author responses that were so thoughtful, and complete). They enhanced the original work by conducting large additional animal studies (funded by Bill & Melinda Gates Foundation-- authors added) and by adding patient data from a second continent (Vietnam, added to existing data from Uganda). They answered my questions in full, as well as those of fellow reviewers. I have no further concerns.

We appreciate this feedback.

Reviewer 4

The concept of using pre-rRNA/rRNA ratios as a surrogate for bacterial killing in murine models is innovative, novel, and of interest. The authors have also supplemented their data with a nice new experiment looking at RS in murine relapse across 4 regimens and 3 treatment durations.

- 1. Critique on RS as an indicator of sterilization vs transcriptional blockade or ATP depletion. "It is not at all surprising that a drug that inhibits RNA polymerase (RIF) and a drug that dramatically depletes the key RNA precursor ATP (BDQ) would slow RNA biosynthesis in M.tb and lead to low rRNA synthesis ratios by virtue of the unstable ETS1 species being cleared rapidly while the stable 23S rRNA lingering. So the finding that RIF and BDQ reduce rRNA synthesis ratios may be coincidental to the drugs' mechanisms of action and have nothing to do with their sterilizing activities"**

The authors make the point that their relapse study favors the argument that RS is measuring sterilizing ability, this point is still not proven. It would be helpful to readers to point out alternative interpretations in the discussion.

We agree that it is important that we emphasize this limitation. We have added the following to lines 474-478.

"we cannot be certain that the association between RS ratio and treatment shortening activity is generalizable to non-rifamycin, non-bedaquiline-based regimens that we have not yet tested. It remains possible that inhibition of RNA or ATP synthesis suppresses the RS ratio in a way that is independent of sterilizing activity."

- 2. Critique on why not test other known sterilizing drugs. Other known sterilizing drugs which do not have mechanisms of action that inhibit RNA synthesis were not tested in the in vitro analysis (Fig.**

3). These include pyrazinamide, moxifloxacin, linezolid, and clofazimine. Disappointingly, the authors make rhetorical arguments here to defend the omission of testing these monotherapies. Their point about PZA is credible, but to make their results as convincing as possible, the others should have been tested. And having tested these other drugs which do not act by the same mechanisms as RIF and BDQ is pertinent to point #1 above. It would be helpful to readers to point out that the RS method was not tested with sterilizing monotherapies such as moxifloxacin, linezolid, and clofazimine somewhere in the discussion.

We now acknowledge this limitation in lines 469-475 as follows.

“A limitation of this report is that we did not test drugs that may have sterilizing activity, including moxifloxacin, clofazimine and linezolid, individually *in vitro* and in mice.”